# Single Laboratory Evaluation of the Q20+ Nanopore Sequencing Kit for Bacterial Outbreak Investigations

**DOI:** 10.3390/ijms252211877

**Published:** 2024-11-05

**Authors:** Maria Hoffmann, Jay Hee Jang, Sandra M. Tallent, Narjol Gonzalez-Escalona

**Affiliations:** Genomics Development and Applications Branch, Division of Food Safety Genomics, Office of Applied Microbiology and Technology, Office of Laboratory Operations and Applied Science, Human Foods Program, Food & Drug Administration, College Park, MD 20740, USA; maria.hoffman@fda.hhs.gov (M.H.); sandra.tallent@fda.hhs.gov (S.M.T.)

**Keywords:** nanopore sequencing, foodborne pathogen, complete genomes, longread sequencing, single laboratory evaluation

## Abstract

Leafy greens are a significant source of produce-related Shiga toxin-producing *Escherichia coli* (STEC) outbreaks in the United States, with agricultural water often implicated as a potential source. Current FDA outbreak detection protocols are time-consuming and rely on sequencing methods performed in costly equipment. This study evaluated the potential of Oxford Nanopore Technologies (ONT) with Q20+ chemistry as a cost-effective, rapid, and accurate method for identifying and clustering foodborne pathogens. The study focuses on assessing whether ONT Q20+ technology could facilitate near real-time pathogen identification, including SNP differences, serotypes, and antimicrobial resistance genes. This pilot study evaluated different combinations of two DNA extraction methods (Maxwell RSC Cultured Cell DNA kit and Monarch high molecular weight extraction kits) and two ONT library preparation protocols (ligation and the rapid barcoding sequencing kit) using five well-characterized strains representing diverse foodborne pathogens. High-quality, closed bacterial genomes were obtained from all combinations of extraction and sequencing kits. However, variations in assembly length and genome completeness were observed, indicating the need for further optimization. In silico analyses demonstrated that Q20+ nanopore sequencing chemistry accurately identified species, genotype, and virulence factors, with comparable results to Illumina sequencing. Phylogenomic clustering showed that ONT assemblies clustered with reference genomes, though some indels and SNP differences were observed, likely due to sequencing and analysis methodologies rather than inherent genetic variation. Additionally, the study evaluated the impact of a change in the sampling rates from 4 kHz (260 bases pair second) to 5 kHz (400 bases pair second), finding no significant difference in sequencing accuracy. This evaluation workflow offers a framework for evaluating novel technologies for use in surveillance and foodborne outbreak investigations. Overall, the evaluation demonstrated the potential of ONT Q20+ nanopore sequencing chemistry to assist in identifying the correct strain during outbreak investigations. However, further research, validation studies, and optimization efforts are needed to address the observed limitations and fully realize the technology’s potential for improving public health outcomes and enabling more efficient responses to foodborne disease threats.

## 1. Introduction

Leafy greens are responsible for nearly half of the produce-related Shiga toxin-producing *Escherichia coli* (STEC) outbreaks in the United States, and recent investigations have implicated agricultural water as a potential source [1]. Currently, the Food and Drug Administration (FDA) outbreak detection protocols for STEC require extensive analysis time (2–4 weeks) [2]. During outbreak investigations, finding the strain that matches the outbreak strain can take several days after isolating each strain from the potential food sources. This delay is crucial, considering that leafy greens have a limited shelf life. During outbreak events and periods of uncertainty, methods are needed to safely preserve these products to prevent unnecessary waste. Therefore, there is a need for methods that can assist in rapidly identifying the correct strain and producing a preliminary sequencing analysis during such investigations, potentially reducing the time required to find the outbreak strain.

Currently, the FDA utilizes Illumina instruments for whole genome sequencing (WGS) analysis of isolated strains for surveillance [3,4,5]. While short-read Illumina MiSeq sequencing technology is highly accurate, it struggles with assembling highly repetitive regions that can span several hundred base pairs [6,7,8,9,10,11]. As a result, the final assembled genome of an outbreak strain typically consists of numerous contigs (e.g., approximately 200 contigs for STECs) and often lacks critical information such as plasmid presence and the location of important genes, including antimicrobial resistance genes (AMR) [10,12,13,14,15]. An alternative approach involves using Oxford Nanopore Technologies (ONT) sequencing, a technology that allows for determining a DNA or RNA sequence by reading DNA or RNA through a biological nanopore and measuring changes in an electrical current that translates into a base call. ONT produces long reads that can identify taxa while sequencing. This method has shown promise in outbreak detection [8,9], but its lower raw read accuracy previously made it unsuitable as a stand-alone technique for certain downstream analyses, such as SNP analysis during outbreak investigations [3,4,16,17,18]. Accurate SNP variant detection is crucial in outbreak investigations, as some strains can differ by only a few SNPs (around 1–10) [3,4,18,19]. However, in 2023, ONT released a new chemistry (Q20+), achieving ≥99% raw read accuracy and high sequencing yield [20,21,22,23,24], making it a potential candidate for improved and faster outbreak response.

Prior to implementation of ONT sequencing as an alternative as an enhanced and faster tool for outbreak response, it is crucial to ensure that the quality obtained is on par with the current gold standard, Illumina sequencing. To achieve this, a pilot study of ONT’s new chemistry, including the V14 DNA library preparation kits and new R10.4.1 flow cell, is necessary to select the optimal combination to perform a more rigorous single laboratory validation (SLV). According to the FDA’s Guidelines for the Validation of Analytical Methods Using Nucleic Acid Sequenced-Based Technologies [25], several important considerations must be taken into account when conducting a verification study for a new change in next-generation sequencing (NGS) chemistry such as the Q20+ from ONT. These considerations include (1) the use of a well-characterized isolate panel for primary NGS/WGS data collection, (2) reproducibility (precision between runs), and (3) repeatability (precision within a run). In this pilot study, only the first consideration will be evaluated, and it will be conducted using a set of reference genomes. Among the parameters to evaluate are (a) average depth of genome coverage, (b) mean read length, (c) sequence length distribution, (d) assembly length, (e) number of contigs, (f) N50, and (g) percentage of recovered open reading frames (ORF) and core genes. Subsequently, in silico tests will be conducted to assess other aspects, such as correct species identification (accuracy, specificity, and sensitivity), genotyping methods [serotyping, AMR, virulence, toxin, multi-locus sequence typing (MLST)], and phylogenomic clustering methods measuring specificity and sensitivity for cluster assignment. A wgMLST approach using the known genome reference of the tested strains as a reference will be employed to determine specificity and sensitivity for determining variants (single nucleotide polymorphisms (SNPS) and/or indels) and the accuracy of the entire tree or specific splits leading to an outbreak lineage. The entire workflow is depicted in Figure 1.

ONT has two main DNA library preparation kits for whole genome sequencing of bacteria: (1) the Ligation Sequencing Kit (SQK-LSK114) and (2) the Rapid Barcoding Sequencing Kit (SQK-RBK11). They differ in the preparation time and steps required to obtain a library ready for sequencing. The RBK kit is faster and less labor-intensive than the LSK kit, albeit producing lower sequencing output. The sequencing performance and output can be highly impacted by the quality of the DNA used for the sequencing. Therefore, we chose to additionally evaluate the performance of these two library preparation kits with two DNA extraction kits that produced DNA with different specifications: (1) the faster and automatized Maxwell RSC Cultured Cell DNA kit (high DNA concentration output but sheared DNA) and (2) the manual Monarch High Molecular Weight (HMW) DNA Extraction Kit (DNA concentration high and with more integrity—longer fragments). For this pilot study, we selected for sequencing five well known and characterized foodborne and healthcare-associated bacterial pathogen strains that pose significant public health threats, belonging to five different species from our collection: *Salmonella enterica* subsp. *enterica*, *Vibrio parahemolyticus*, *Shigella sonnei*, *Escherichia coli*, and *Klebsiella pneumoniae* (Table 1). These organisms are closely monitored by health services due to their roles in outbreaks, potential for severe illness, and emerging antibiotic resistance. The DNA was extracted from overnight cultures with the two different kits and sequenced using the two ONT library preparation kits separately, for a total of four runs, as shown in Table 2 and Appendix A.

## 2. Results

Preliminary testing of the Q20+ chemistry was done to establish which sequencing kit and DNA combination could be used for easily and completely closing bacterial genomes for later validation efforts. As shown in Table 2, each combination resulted in different total output and total read numbers. The highest output both in read numbers and sequence was achieved with the LSK114/Maxwell combination (10.65 Gb and 1.76 M reads). The lowest was obtained with the RBK114/NEB combination (3.28 Gb and 305k reads). The higher N50 length reads per strain per run were observed on the runs using the DNA extracted with the NEB HMW kit with any of the two DNA library combinations (Appendix A). The higher number of reads per strain per run were observed on the runs using the Maxwell/LSK114 combination. This partially explains why the RBK114/NEB run had the lowest output.

### 2.1. Genome Sequencing QC

The data generated from the four runs were compared for assembly statistics: (a) assembly length (bp), (b) average depth of genome coverage, (c) genome completeness, (d) number of contigs, and (e) % ORF recovered. Table 3 shows the assembly length for each strain per run and the average depth coverage for each genome per run. The assembly length per strain per run showed great variation. For the *Salmonella* strain, the assembled genome length varied from 2 kb less to 5 kb more compared to its reference genome; for the *Vibrio parahemolyticus*, it varied from 5 kb to 38 kb more compared to the reference genome; for the *Shigella sonnei* it varied from 17 kb less to 21 kb more compared to its reference genome; for the *E. coli* O157:H7, it varied from 2 kb less to 22 kb more compared to its reference genome; and for the *Klebsiella pneumoniae*, the variation was less pronounced, from 0.2 kb to 3.8 kb more compared to its reference genome. These variations do not appear to be linked to the sequencing depth, since, for example, *Klebsiella pneumoniae* showed the higher variation for the highest coverage (>300×).

Regarding genome completeness, most genomes were complete except for CFSAN076620 (*E. coli* O157:H7, run RBK114-NEB and LSK114-NEB) and CFSAN086181 (*Klebsiella pneumoniae*, run RBK114-NEB) (Table 4). Most of the ORFs (>99%) were recovered by ONT Q20+ in all of the combinations compared to the reference genome for each strain (Table 5). Based on the analyzed genome and sequencing combinations, there were still issues with indels, and the recovery of some ORFs was not possible. The results were not identical and showed inconsistencies between runs.

### 2.2. In Silico Analyses

The data generated from the four runs were compared for diverse in silico tests or assays: (a) correct species identification (accuracy, specificity, and sensitivity), (b) genotyping methods (serotyping, AMR, virulence, toxin, MLST), and (c) phylogenomic clustering methods measuring the specificity and sensitivity for cluster assignment.

(a)Correct species identification (accuracy, specificity, and sensitivity). Two different methods for species identification were used. The initial identification was performed using the WIMP workflow, and each species was identified correctly. This initial assessment was confirmed using a different taxonomic tool (Kraken 2) using the final assemblies for each sample. The assemblies for each strain from each run were run through it, and each species was identified correctly (100% accuracy, specificity, and sensitivity).(b)Genotyping methods (serotyping, AMR, virulence, toxin, MLST). Regarding the serotyping, only two serotyping schemes were available for the five different species included in this pilot study (*Salmonella* and *E. coli*). There was a 100% agreement with the serotype predictions, as all *Salmonella* and *E. coli* ONT assemblies were identified as serotype Bareilly and O157:H7, respectively. Similarly, in silico sequence type (ST) determination yielded identical results for ONT assembly (ST-909 for CFSAN000189, ST-635 for CFSAN123154, ST-152 for CFSAN030807, ST-11 for CFSAN076620, and ST-629 for CFSAN086181), except for one assembly (CFSAN000189) that showed an incomplete MLST profile due to an indel in the *sucA* gene (NEB-LSK114). All ONT assemblies for each strain agreed 100% for the presence of specific antimicrobial resistance (AMR) genes compared to the reference genome for each strain. CFSAN000189 carried a single copy of the *aac(6′)-Iaa* gene, CFSAN123154 carried a single AMR gene (*blaCARB*-36), CFSAN076620 carried two copies of the *catA1* gene, and CFSAN086181 carried five genes (*blaSHV-110, blaSHV-81, fosA6, OqxB*, and *OqxA*). An issue arose with the copy number for the AMR genes for strain CFSAN030807, which carried four AMR genes [*aph*(6)-Id, *aph*(3″)-Ib, *sul2*, and *tet*(A)]. However, in the assemblies for runs M-LSK114 and M-RBK114, the plasmid carrying the AMR genes was double in size compared to the original plasmid, likely due to an assembly issue (we have observed this same issue in many Q20+ ONT assemblies), resulting in the same sequence being concatenated twice.

For this pilot study, in silico analyses were conducted on ONT assemblies of the five strains to determine the presence of virulence and toxin genes. The genes reported for each species in the virulence factors database (VFDB) task template included in the Ridom Seqsphere software v9.0.8 were utilized for these analyses, except for *E. coli*, where Virulence Finder at DTU was used instead. The reference genome of *Salmonella enterica* subsp. *enterica* (CFSAN000189) Bareilly contained 105 virulence genes, all of which were consistently identified in ONT assemblies. Similarly, the *Vibrio parahemolyticus* (CFSAN123154) strain, characterized as *tdh*−/*trh*+, contained 42 virulence genes in its reference genome, all of which were also found in every ONT assembly. Additionally, the *Shigella sonnei* (CFSAN030807) strain’s reference genome contained 10 virulence genes, all of which were consistently identified in ONT assemblies. This strain has lost a larger plasmid that carried 42 additional virulence genes compared to the current genome available at NCBI. In the case of *E. coli* (CFSAN076620), the reference genome contained 20 relevant virulence genes, which were consistently detected across different ONT combinations, with only one gene (*toxB*) missing in certain ONT assemblies, specifically NEB-RBK114 and NEB-LSK114. Finally, the reference genome of *Klebsiella pneumoniae* (CFSAN086181) harbored 52 virulence genes, all of which were present in every ONT assembly.

(c)Phylogenomic clustering methods measuring the specificity and sensitivity for cluster assignment. For this last step, a wgMLST approach was used to determine specificity (using as a reference genome the known genome sequence of the tested strains) and sensitivity for determining variants (SNPS and/or indels), accuracy of entire tree, or specific splits (e.g., leading to an outbreak lineage).

### 2.3. wgMLST Analyses for CFSAN000189 (Salmonella Bareilly)

For this step, two different wgMLST analyses were performed: (1) against a collection of closely related *Salmonella* Bareilly genomes and (2) against the reference genome for that same strain (obtained by PacBio or by a hybrid assembly of ONT and Illumina data). Total genes used in the wgMLST were 4312 loci. Of those, 3991 loci (93%) were identical among all samples (including the PacBio and hybrid genome). The ONT assemblies obtained with the Q20+ chemistry (V14) by either combination clustered with the corresponding genome in a SNP NJ tree after a wgMLST analysis (Figure 2A), albeit with at least 100 to 124 SNPs and 30–39 indels, compared to the reference genome (Figure 2B and Table 6). The SNP differences matrix is shown in Table 6. We must add that this strain was sequenced using the same initial bacterial culture; thus, the DNA must be the same in terms of methylations. The complete NJ SNP tree can be found on Appendix A. Additionally, the Illumina sequences [obtained using a MiSeq (M)] assembled were also included in the wgMLST analysis and were indistinguishable from the PacBio or ONT hybrid assemblies.

### 2.4. wgMLST Analyses for CFSAN123154 (Vibrio parahemolyticus)

Similar to the approach used for *Salmonella*, two different wgMLST analyses were performed for the *Vibrio parahemolyticus* strain: (1) against a collection of closely related *Vibrio parahaemolyticus* genomes and (2) against the reference genome for that same strain (obtained by PacBio). The total genes used in the wgMLST were 4440 loci. Of those, 4078 loci (92%) were identical among all samples (same DNA prepared with different library kit and run in different flow cells). The ONT assemblies obtained with the Q20+ chemistry (V14) by either combination clustered with the corresponding genome in an SNP NJ tree after a wgMLST analysis (Figure 3A), albeit with at least 100 to 133 SNPs and 53 to 81 indels, compared to the reference genome (Figure 3B and Table 7). The complete NJ SNP tree can be found on Appendix A. The SNP differences matrix is shown in Table 7.

### 2.5. wgMLST Analyses for CFSAN030807 (Shigella sonnei)

Similar to the approach used for *Salmonella*, two different wgMLST analyses were performed for the *Shigella sonnei* strain: (1) against a collection of closely related *Shigella sonnei* genomes (ST-152) and (2) against the reference genome for that same strain (obtained by PacBio). Total genes used in the wgMLST were 3699 loci. Of those, 3657 loci (99%) were identical among all samples (same DNA prepared with different library kit and run in different flow cells). The ONT assemblies obtained with the Q20+ chemistry (V14) by either combination were clustered with the corresponding genome in a SNP NJ tree after a wgMLST analysis (Figure 4A), with at least 0 to 2 SNPs and 1 to 13 indels, compared to the reference genome (Figure 4B and Table 8). The SNP differences matrix is shown in Table 8. We must add that this strain was sequenced using the same initial bacterial culture, so the DNA must be the same in terms of methylations. The complete NJ SNP tree can be found in Appendix A.

### 2.6. wgMLST Analyses for CFSAN076620 (E. coli O157:H7)

Similar to the approach used for *Salmonella*, two different wgMLST analyses were performed for the *E. coli* strain: (1) against a collection of closely related *E. coli O157:H7* genomes (ST-11) and (2) against the reference genome for that same strain (obtained by PacBio). Total genes used in the wgMLST were 4639 loci. Of those, 4500 loci (97%) were identical among all samples (same DNA prepared with different library kits and run in different flow cells). The ONT assemblies obtained with the Q20+ chemistry (V14) by either combination clustered with the corresponding genome in a SNP NJ tree after a wgMLST analysis (Figure 5A), albeit with at least 3 to 28 SNPs and 32 to 74 indels, were compared to the reference genome (Figure 5B and Table 9). The SNP differences matrix is shown in Table 9. We must add that this strain was sequenced using the same initial bacterial culture, so the DNA must be the same in terms of methylations.

### 2.7. wgMLST Analyses for CFSAN086181 (Klebsiella pneumoniae)

Similar to the approach used for *Salmonella*, two different wgMLST analyses were performed for the *Klebsiella pneumoniae* strain: (1) against a collection of closely related *Klebsiella pneumoniae* genomes and (2) against the reference genome for that same strain (obtained by PacBio). Total genes used in the wgMLST were 4841 loci. Of those, 4778 loci (98.7%) were identical among all samples (same DNA prepared with different library kit and run in different flow cells). The ONT assemblies obtained with the Q20+ chemistry (V14) by either combination clustered with the corresponding genome in a SNP NJ tree after a wgMLST analysis (Figure 6A), with one to nine SNPs and 10 to 24 indels compared to the reference genome (Figure 6B and Table 4). The SNP differences matrix is shown in Table 10. Note that this strain was sequenced using the same initial bacterial culture, so the DNA should be identical in terms of methylations. The complete NJ SNP tree can be found in Appendix A.

### 2.8. Testing the New Software Update (MinKNOW 23.04.5) for the Same Samples Using the RBK-114 Kit

The new software update (MinKNOW 23.04.5) was tested using the RBK-114 kit on the same samples. During the evaluation of the new Q20+ chemistry, ONT released an update for the MinKNOW software (v23.04.5), which incorporated a new sampling rate (5 kHz) and basecaller model (dna_r10.4.1_e8.2_5khz_400bps_sup@v4.2.0). ONT claimed that there were no differences in the accuracy of the SUP 400 bps (5 kHz) compared to the previous SUP at 260 bps (4 kHz). A test using the same five DNA strains and the rapid barcoding kit (RBK-114) was performed. The output was very similar to the observed RBK runs (741.66k reads and 3.9 Gb, flow cell number FAW81132). The procedure for data analysis remained consistent, employing two different wgMLST analyses for each strain: (1) against a collection of closely related genomes and (2) against the reference genome for the same strain. Every strain clustered in the same cluster for each individual species, as observed with the SUP 260 bps. Regarding indels and SNPs, 

-CFSAN000189 had 32 SNPs and 56 indels compared to its reference genome,-CFSAN123154 had 77 SNPs and 78 indels compared to its reference genome,-CFSAN030807 had 0 SNPs and 5 indels compared to its reference genome,-CFSAN076620 had 3 SNPs and 27 indels compared to its reference genome, and-CFSAN086181 had 2 SNPs and 7 indels compared to its reference genome.

## 3. Discussion

This study was conducted to address the limitations of current FDA pathogen sequencing methods, which are time consuming and require expensive equipment. One possible solution is the use of less expensive sequencers, which could increase access to this type of technology for many potential users. Expanding access to this type of technology is essential to enhance the effectiveness and speed of foodborne outbreak responses. This research aimed to evaluate whether Oxford Nanopore Technologies (ONT) MinION with Q20+ chemistry can offer rapid and accurate identification of foodborne pathogens, including SNP differences, serotypes, and antimicrobial resistance genes. The goal was to assess if ONT Q20+ technology could provide near real-time pathogen identification, potentially overcoming the drawbacks of current methodologies. The study aimed to bridge advanced genomic technologies with regulatory science, enabling more effective responses to foodborne disease threats and enhancing public health outcomes.

To achieve this, the study evaluated ONT Q20+ chemistry using five well-characterized strains representing diverse foodborne pathogen traits. Various DNA extraction (Maxwell and NEB HMW) and library preparation methods (RBK-114 and LSK-114) were compared to optimize sequencing outcomes. Any combination proved sufficient to generate high-quality bacterial genomes using the Flye assembler as observed by numerous authors under similar conditions [20,21,22]. However, the ligation kit (LSK-114) produced a higher output and longer reads when used in combination with the NEB HMW DNA extraction kit, as observed previously by other authors [26,27]. If you want to assemble very complex bacterial genomes, the combination of high molecular weight DNA extraction (less sheared DNA) and the ligation sequencing kit (SQK-LSK114) is recommended as observed for STECs [28,29]. For further validation studies, the combination of the rapid library sequencing kit (RBK-114) and the automated DNA extraction kit (Maxwell RSC Cultured Cell DNA kit) proved to be optimal and preferred because of speed and reduced errors during DNA extraction and sequencing preparations resulting in less laboratory discrepancies. Variations in assembly length and genome completeness were observed across different combinations of extraction and sequencing kits. The differences observed in genome completeness and assembly length for the STEC and *Klebsiella pneumoniae* are likely due to variations in the downstream assembly process rather than the DNA extraction or library preparation methods. Given that both methods consistently achieved sufficient coverage (above 30X), we do not expect the extraction or preparation methods to significantly impact genome quality, as the primary difference between the methods relates to total read yield rather than assembly accuracy. Most genomes achieved high completeness (>99% ORF recovery) with ONT Q20+, as has been observed previously [22,23,24], but some variations in assembly length and missing ORFs were noted, suggesting the need for further optimization. In addition, duplication of small plasmids was observed in some cases when using the Flye assembler, as observed by others [21,30,31,32], specifically for strain CFSAN030807, where the expected small plasmid of 8401 bp carrying four AMR genes was 16,802 bp in size for the assemblies of M—RBK114 and M—LSK114. Therefore, the reason the plasmid is double in size seems random, or perhaps the DNA extracted with that specific kit created smaller fragments for that plasmid that were randomly misassembled due to some minor SNP differences.

Regarding the in silico analyses, the Q20+ nanopore sequencing chemistry demonstrated 100% for accurate species identification, genotyping (including serotyping and AMR gene detection), and virulence factor identification comparable to Illumina sequencing. Phylogenomic clustering methods (wgMLST) showed that ONT assemblies clustered with reference genomes, although some indels and SNP differences were observed with some bacterial species and less pronounced with others. Supplementing with Illumina data helped resolve these differences. Other authors have similar results for other enterobacterial species (*Citrobacter freundii*, *Klebsiella pneumoniae*, *Enterobacter hormaechei*, *Enterobacter cloacae*, and *Escherichia coli*) and noticed that there was still a need to supplement with Illumina data in order to correct those indels or SNPs [21]. Despite these variations, the overall clustering and similarity to reference genomes highlight the robustness of nanopore sequencing results for assisting in detecting the correct strain during outbreak investigations. While SNP variation is important, genomes within an outbreak may differ by a few SNPs due to natural mutation rates during strain culturing. Therefore, integrating genetic data with epidemiological context is essential for accurately interpreting phylogenetic relationships and assessing potential outbreak inclusion, ultimately enhancing our ability to draw informed conclusions. The SNPs observed were randomly distributed in the genome of some of the analyzed strains. This addition clarifies that any observed SNP differences in the phylogenetic analyses likely stemmed from sequencing and analysis processes rather than genetic variations in the sampled bacteria. It is important to note that these analyses were conducted using DNA from the same initial bacterial culture, and the impact of methylations on the basecallers should have been the same for each strain and should not reflect on the SNPs observed for each run for that specific strain (Appendix A).

The assembled genomes from each combination were compared against their reference genome annotated coding sequences (CDS) using a highly conservative requirement for a 100% match, as described by Sanderson et al. (2024) [22]. Both indels and SNP variants were identified using this process. Regulatory actions demand a high level of certainty to identify an outbreak strain. Even though we performed a single run per combination (except for the Maxell RBK-114 combination), we believe that it is equivalent to multiple replicates of the same sample. Although only a single run was performed for each combination (except for the Maxell RBK-114 combination), this is considered equivalent to multiple replicates of the same sample. Since the same bacterial culture and chemistry were used, with only variations in library preparation kits and procedures, the final sequence data quality should remain unaffected. As mentioned by Sanderson et al. 2024 [22], factors such as pore number, operator skill, and library type can influence read output and yield. Additionally, while only five strains from different species were analyzed, Sanderson et al. (2024) cautioned that these findings may not be broadly applicable across all strains or species. Most studies evaluating the newer ONT chemistry focus on a single species or a single sequencing run, limiting the scope of the technology’s assessment [20,23,33,34]. In contrast, this study expands the evaluation by testing five different bacterial species and performing five sequencing run replicates from the same initial culture. This approach enhances the analytical robustness of the study, allowing for a more comprehensive evaluation. Additionally, it assesses sequencing errors to determine whether they consistently occur at the same genomic positions—potentially linked to methylation patterns—or if they are randomly distributed as shown in this study. Our results also demonstrated that assembly accuracy varied by species/strain compared to the reference (Table 6, Table 7, Table 8, Table 9 and Table 10). A key innovation of this study is that it demonstrates the potential of ONT newer chemistry to deliver more accurate sequences than the previous ONT chemistry (R9.4.1), faster and at a lower initial cost, making it a more efficient tool for bacterial screening during outbreak investigations.

During the initial evaluation of the Q20+ chemistry, the study coincided with a major update to the basecaller. Using the RBK-114 kit, the same five DNA strains were processed with both the new and old basecaller models. ONT claimed no difference in accuracy between the SUP 400 bps (5 kHz) and the previous SUP 260 bps (4 kHz) models, and our findings largely supported this. Additionally, downstream analyses showed that clustering for all strains was consistent with the results obtained using the SUP 260 bps model, indicating that the new basecaller model did not affect overall clustering or comparative accuracy in detecting SNPs and indels. Overall, testing confirmed that the update to MinKNOW 23.04.5 and its new basecaller model maintains accuracy and performs comparably to the previous version, validating ONT’s claims. This suggests that users can confidently transition to the new software update without concerns over data quality loss. However, it should be noted that ONT has made several basecaller changes since the start of the evaluation, which could raise concerns for both current and potential new users.

## 4. Materials and Methods

### 4.1. Bacterial Strains and Media

The bacterial strains used in this study belonged to 5 different species associated with foodborne outbreak or cases collected by the Center for Food Safety and Applied Nutrition (CFSAN) as part of our Genome Trakr surveillance program (Table 1). These 5 strains were selected to represent various species, antimicrobial resistance (AMR) genes, common foodborne pathogen, diverse % GC content (medium and high), SNP recovery as part of the same outbreak, toxin gene detection, and diverse plasmid content. Bacterial strains were cultured in brain heart infusion (BHI) broth (Thermo Fisher Scientific, Waltham, MA, USA) and stored in BHI broth with 20% glycerol stocks at −80 °C.

### 4.2. DNA Extraction and Quality and Quantification Measurement

Genomic DNA from each strain was extracted using either the Maxwell RSC Cultured Cell DNA Kit with a Maxwell RSC Instrument (Promega Corporation, Madison, WI, USA) according to the manufacturer’s instructions for Gram-negative bacteria with additional RNase treatment or the Monarch HMW DNA Extraction kit (New England Biolabs, Ipswich, MA, USA). DNA concentration was determined by a Qubit 4 Fluorometer (Invitrogen, Carlsbad, CA, USA) according to the manufacturer’s instructions. The DNA integrity was determined using a Femto Pulse System (Agilent, Santa Clara, CA, USA).

### 4.3. Whole Genome Sequencing and Assembly Using Long Reads

DNA extracted from each strain was sequenced using the GridION Sequencing Device Mk1 (Oxford Nanopore Technologies, Oxford, UK) in different conformations (Table 2). The sequencing libraries were prepared using different protocols: (1) Rapid Barcoding Kit 24 V14 (SQK-RBK114.24) or (2) Ligation Sequencing Kit v14 (SQK-LSK114) with Native Barcoding Kit (EXP-NBD114). They were then run in FLO-MIN114 (R10.4.1) flow cells, according to the manufacturer’s instructions, for 48 h (Oxford Nanopore Technologies). The sequencing run was live base called using different MinKNOW software versions (see Table 2) using the super-accurate basecalling model (SUP, basecall_model—dna_r10.4.1_e8.2_260bps_sup@v3.5.2). The reads < 2000 bp and quality scores of <10 were discarded for downstream analysis using Fitlong v0.2.1 (https://github.com/rrwick/Filtlong, accessed on 30 October 2024) with default parameters. The genomes for each strain were obtained by de novo assembly using the Flye program v2.9.2 [35], using the following parameters: --nano-hq, --read-error 0.03, and -i 4. Flow cell FAW81132 was run using a different sampling rate (5 kHz) and basecalling method: dna_r10.4.1_e8.2_5khz_400bps_sup@v4.2.0.

The short read whole genome sequences for these strains were generated by Illumina MiSeq sequencing with the MiSeq V3 kit using 2 × 250 bp paired-end chemistry (Illumina, San Diego, CA, USA), according to the manufacturer’s instructions, at 100X coverage. The libraries were constructed using 100 ng of genomic DNA using the Illumina^®^ DNA Prep (M) Tagmentation (Illumina, San Diego, CA, USA), according to the manufacturer’s instructions. Reads were trimmed with Trimmomatic v0.36 [36] using default parameters.

### 4.4. Hybrid Assembly

The complete genome for each strain from run FAW25027 was obtained by a de novo hybrid assembly using both nanopore and MiSeq data for that strain with Unicycler v0.5.0 [37]. Unicycler assembled the chromosome and plasmids as circular closed and oriented the chromosome to start at the *dnaA* gene.

### 4.5. Species Identification

The initial species identification was performed using the reads as they were being produced by the sequencing device for every single sample and analyzed using the ONT “What’s in my pot” (WIMP) workflow contained in the EPI2ME cloud service (Oxford Nanopore Technologies) after 5 h. The final taxonomical classification and species confirmation were performed using Kraken 2 [38] implemented in the GalaxyTrakr website [39] using the default database.

### 4.6. In Silico MLST and Serotyping

The initial analysis and identification of the strains were performed using an in silico MLST approach using Ridom SeqSphere+ v9.0.8 (Ridom, Münster, Germany) using their MLST task template for each individual species as follows: for *Salmonella enterica* (https://enterobase.warwick.ac.uk/species/index/senterica, all accessed on 30 October 2024), *Vibrio parahemolyticus* (https://pubmlst.org/organisms/vibrio-parahaemolyticus), *Shigella sonnei* (https://enterobase.warwick.ac.uk/species/index/ecoli), *Escherichia coli* (https://enterobase.warwick.ac.uk/species/index/ecoli), and *Klebsiella pneumoniae* (https://bigsdb.pasteur.fr/klebsiella/) (*Salmonella* and *E. coli*). Only two serotyping schemes were available for the 5 different species included in this pilot study (*Salmonella* and *E. coli*). For *Salmonella enterica* we used SeqSero2 v1.1.0 (http://www.denglab.info/SeqSero2), and for *E. coli* we used SerotypeFinder 2.0 (https://cge.food.dtu.dk/services/SerotypeFinder/) [40].

### 4.7. In Silico AMR Gene Detection

The AMR gene detection was performed using ResFinder v4.5.0 (http://genepi.food.dtu.dk/resfinder) [41].

### 4.8. In Silico Identification of Virulence Genes

The de novo assemblies were batch screened for virulence genes using the Ridom SeqSphere+ task template for the virulence genes reported for each species in the virulence factors database (VFDB—http://www.mgc.ac.cn/VFs/main.htm), except for *E. coli*, where Virulence Finder at DTU was used instead.

### 4.9. Phylogenetic Relationship of the Strains Determined by wgMLST Analysis

The phylogenetic relationship of the strains was assessed by a whole genome multilocus sequence typing (wgMLST) analysis using Ridom SeqSphere+ v9.0.8. A separate wgMLST was created for each individual species using the genome available at NCBI or obtained in house for that same strain as reference (Table 1). Genes that were repeated in more than one copy in any of the two genomes were removed from the analysis as failed genes. A task template was then created that contains both core and accessory genes for each reference strain for any future testing. Each individual locus (core or accessory genes) from the reference strain was assigned allele number 1. The assemblies for each individual strain closed genome in this study were queried against the task template for each individual species/strain. If the locus was found and was different from the reference genome or any other queried genome already in the database, a new number was assigned to that locus and so on. After eliminating any loci that were missing or found containing indels (showed as failed call in the software) from the genome of any strain used in our analyses, we performed the wgMLST analysis. These remaining loci were considered the core genome shared by the analyzed strains. We used Nei’s DNA distance method [42] to calculate the matrix of genetic distance, considering only the number of same/different alleles in the core genes. A neighbor-joining (NJ) tree was built using pairwise ignoring missing values and the appropriate genetic distances after the wgMLST analysis. wgMLST uses the alleles number of each locus to determine the genetic distance and build the phylogenetic tree. The use of allele numbers reduces the influence of recombination in the dataset studied and allows for fast clustering determination of genomes.

### 4.10. Nucleotide Sequence Accession Numbers

The raw ONT data for each individual sample were deposited in GenBank under BioProject accession number PRJNA1134675. Each sample SRA number matching its corresponding sample can be found in Appendix A.

## 5. Conclusions

The evaluation of the Q20+ nanopore sequencing chemistry in this single laboratory evaluation study demonstrates its potential for assisting in finding the correct strain during outbreak investigations. Standardizing the extraction and sequencing protocols is crucial for optimizing workflow and ensuring reproducibility and accuracy across different strains and outbreak scenarios. While the study encountered challenges in achieving consistent and reliable results with ONT Q20+ chemistry, the findings provide valuable insights into the technology’s potential for enhancing foodborne pathogen detection and surveillance. Continued research and optimization efforts are necessary to address the observed limitations and fully realize the impact of nanopore sequencing on public health and regulatory science. Further verification studies and optimizations are essential to integrate nanopore sequencing into routine outbreak surveillance and response protocols. Overcoming current limitations will be crucial for incorporating these technologies into routine foodborne pathogen surveillance, ultimately leading to improved public health outcomes and more agile responses to foodborne disease threats. The accuracy of SNPs in the assemblies compared to Illumina or PacBio assemblies remains an issue. This is being addressed in future iterations of the ONT basecaller, but the improvements need to be properly assessed and verified.

## Figures and Tables

**Figure 1 ijms-25-11877-f001:**
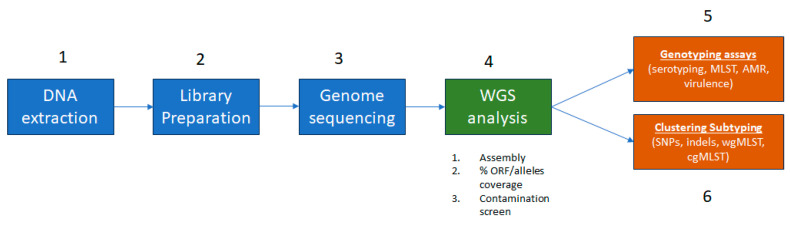
NGS validation workflow for pure bacterial isolates according to the Guidelines for the Validation of Analytical Methods Using Nucleic Acid Sequenced-Based Technologies from the FDA (https://www.fda.gov/food/laboratory-methods-food/foods-program-methods-validation-processes-and-guidelines, accessed on 30 October 2024).

**Figure 2 ijms-25-11877-f002:**
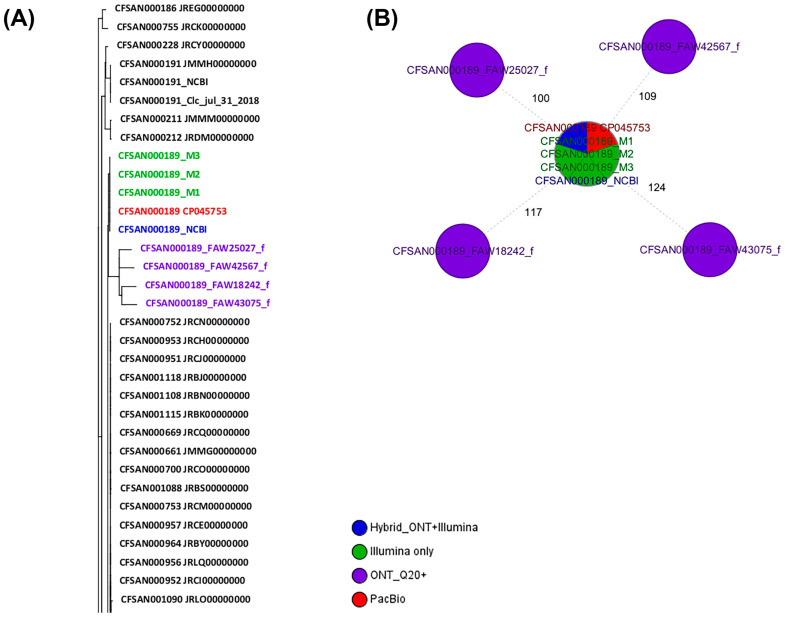
Results of the wgMLST analyses for CFSAN000189. (**A**) Snapshot of the NJ tree generated from the wgMLST analysis of the ONT assemblies obtained by the different combinations tested against a set of known genomes closely related to that same strain (ST909). (**B**) Minimum spanning tree (MST) showing the differences between the different CFSAN000189 assemblies obtained by different sequencing technologies. The complete NJ tree can be found in Appendix A.

**Figure 3 ijms-25-11877-f003:**
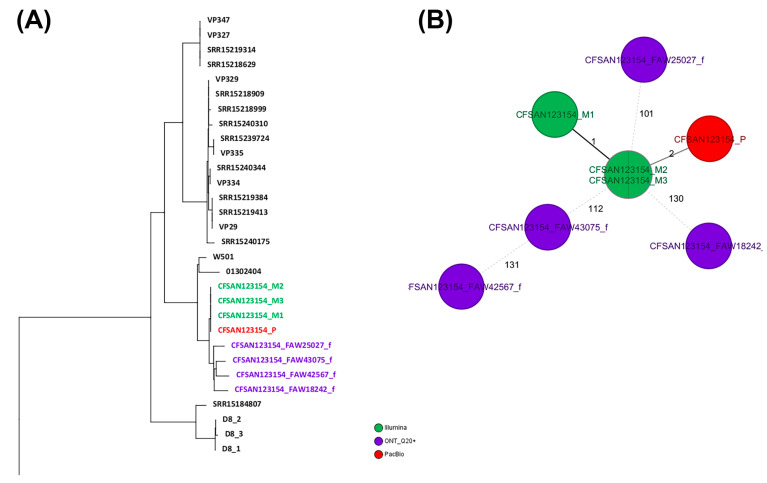
Results of the wgMLST analyses for CFSAN123154. (**A**) NJ tree generated from the wgMLST analysis of the ONT assemblies obtained by the different combinations tested against a set of known genomes closely related to that same strain. (**B**) Minimum spanning tree (MST) showing the differences between the different CFSAN123154 assemblies obtained by different sequencing technologies. The complete NJ tree can be found in Appendix A.

**Figure 4 ijms-25-11877-f004:**
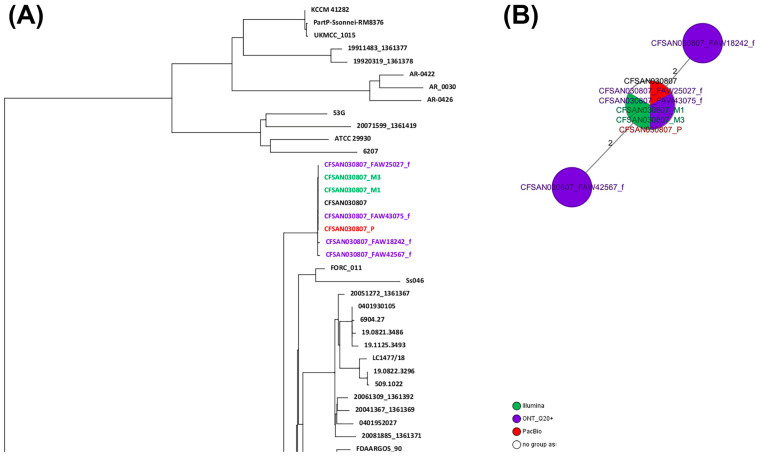
Results of the wgMLST analyses for CFSAN030807. (**A**) Snapshot of the NJ tree generated from the wgMLST analysis of the ONT assemblies obtained by the different combinations tested against a set of known genomes (156) closely related to that same strain (ST152). (**B**) Minimum spanning tree (MST) showing the differences between the different CFSAN030807 assemblies obtained by different sequencing technologies. The complete NJ tree can be found in Appendix A.

**Figure 5 ijms-25-11877-f005:**
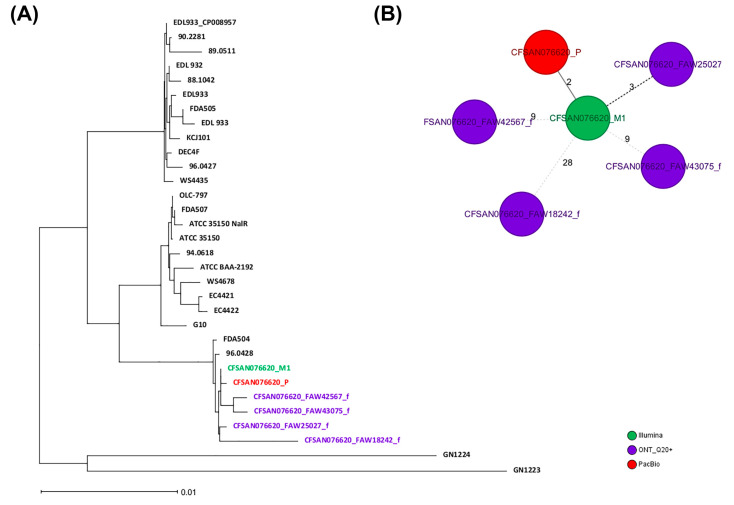
Results of the wgMLST analyses for CFSAN076620. (**A**) NJ tree generated from the wgMLST analysis of the ONT assemblies obtained by the different combinations tested against a set of known genomes (54) closely related to that same strain (ST629). (**B**) Minimum spanning tree (MST) showing the differences between the different CFSAN076620 assemblies obtained by different sequencing technologies.

**Figure 6 ijms-25-11877-f006:**
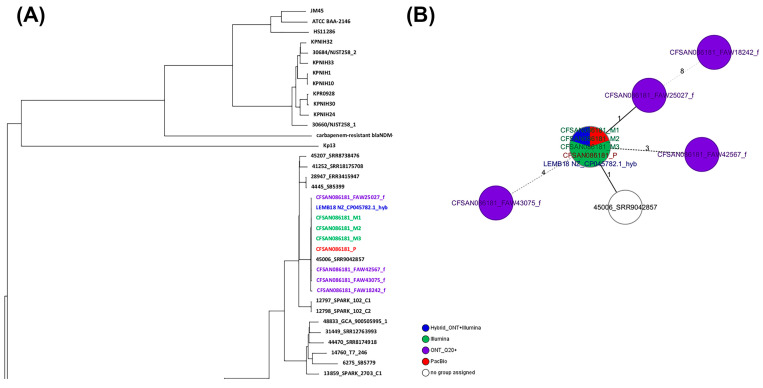
Results of the wgMLST analyses for CFSAN086181. (**A**) Snapshot of the NJ tree generated from the wgMLST analysis of the ONT assemblies obtained by the different combination tested against a set of known genomes (54) closely related to that same strain (ST629). (**B**) Minimum spanning tree (MST) showing the differences between the different CFSAN086181 assemblies obtained by different sequencing technologies. The complete NJ tree can be found in Appendix A.

**Table 1 ijms-25-11877-t001:** Verification strains set used in this single laboratory evaluation study.

CFSAN	Organism	Serovar	Accession No. ^a^
CFSAN000189	*Salmonella enterica* subsp. *enterica*	Bareilly	NZ_CP034177.1, NZ_CP034178.1
CFSAN123154	*Vibrio parahemolyticus*		Our own data
CFSAN030807	*Shigella sonnei*		NZ_CP023645.1, NZ_CP023646.1, NZ_CP023647.1, NZ_CP023648.1, NZ_CP023649.1, NZ_CP023650.1, NZ_CP023651.1, NZ_CP023652.1, NZ_CP023653.1
CFSAN076620	*Escherichia coli*	O157:H7	Our own data
CFSAN086181	*Klebsiella pneumoniae*		NZ_CP045782

^a^ at NCBI.

**Table 2 ijms-25-11877-t002:** Initial testing of the Q20+ chemistry (RBK114 or LSK114 with R10.4.1 flow cells).

Runs	Flowcell No.	DNA Extraction Kit	Library Prep Kit ^a^	Basecalling ^b^	Output	Read No.
1	FAW43075	Maxwell	SQK-LSK114	SUP 260 bps	10.65 Gb	1.76 M
2	FAW25027	Maxwell	SQK-RBK114.24	SUP 260 bps	4.3 Gb	705.61k
3	FAW42567	NEB	SQK-LSK114	SUP 260 bps	8.83 Gb	879.33k
4	FAW18242	NEB	SQK-RBK114.24	SUP 260 bps	3.28 Gb	305.31k

Maxwell—Maxwell RSC Cultured Cell DNA kit, NEB—Monarch high molecular weight (HMW). ^a^ We used 50 ng per strain per flow cell in the case of the RBK kit and 400 ng for the LSK, as recommended by the manufacturer’s protocols for each kit. ^b^ SUP—Super-accurate, MinKNOW software version 22.12.5, basecall_model—dna_r10.4.1_e8.2_260bps_sup@v3.5.2.

**Table 3 ijms-25-11877-t003:** Assembly statistics showing genome size and coverage for each sample sequenced using any of the four combinations tested.

			Total Assembly Length (bp)				Coverage (X)		
Sample	Ref. Genome Size (bp)	FAW25027	FAW18242	FAW43075	FAW42567	FAW25027	FAW18242	FAW43075	FAW42567
^a^ CFSAN000189	4,808,521	4,806,592	4,806,618	4,813,526	4,806,598	92	145	357	305
^b^ CFSAN123154	5,021,303	5,043,592	5,069,096	5,053,959	5,037,119	58	41	174	405
^c^ CFSAN030807	4,848,280	4,861,924	4,869,410	4,866,704	4,831,502	210	63	329	180
^d^ CFSAN076620	5,443,903	5,465,887	5,442,156	5,465,274	5,445,884	42	50	277	141
^e^ CFSAN086181	5,221,690	5,221,905	5,222,423	5,225,442	5,221,867	79	71	344	184

CFSAN000189 (*Salmonella* Bareilly), CFSAN123154 (*Vibrio parahemolyticus*), CFSAN030807 (*Shigella sonnei*), CFSAN076620 (*E. coli* O157:H7), and CFSAN086181 (*Klebsiella pneumoniae*). Genome references at NCBI if available: ^a^ CP045753.1 and CP045752.1, ^b^ our own data, ^c^ GCA_002442535.1, ^d^ our own data, and ^e^ NZ_CP045782.

**Table 4 ijms-25-11877-t004:** Assembly statistics showing genome completeness and contig number for each sample sequenced using any of the four combinations tested.

		Genome Completeness				Contig Number		
Sample	FAW25027	FAW18242	FAW43075	FAW42567	FAW43075	FAW42567	FAW25027	FAW18242
^a^ CFSAN000189	Y	Y	Y	Y	2	2	2	2
^b^ CFSAN123154	Y	Y	Y	Y	5	3	4	6
^c^ CFSAN030807	Y	Y	Y	Y	6	6	6	7
^d^ CFSAN076620	Y	N	Y	N	3	2	3	8
^e^ CFSAN086181	Y	N	Y	Y	2	1	1	1

Y means completely closed and N means incomplete chromosome as reported by Flye. Genome references at NCBI if available: ^a^ CP045753.1 and CP045752.1, ^b^ TBD, ^c^ GCA_002442535.1, ^d^ TBD, and ^e^ NZ_CP045782.1. Expected contig number: *Salm* (2), *Vp* (4), *Shig* (8), *E. coli* (2), and *Kleb* (1).

**Table 5 ijms-25-11877-t005:** Percentage of ORF recovered using any of the testing combinations for each strain per run. Indels and missing genes are also shown.

	% ORF Genes Recovered Intact Using wgMLST Scheme (Genes Missing/Indels)				
Samples	FAW43075	FAW25027	FAW42567	FAW18242	Total Genes ^a^
CFSAN000189	99.2 (1/30)	99.1 (2/37)	99.1 (1/36)	99.3 (0/30)	4296
CFSAN123154	99.5 (2/52)	99.2 (5/57)	99.5 (9/44)	98.9 (2/79)	4440
CFSAN030807	99.8 (2/8)	100 (1/0)	99.8 (1/6)	99.7 (2/11)	3669
CFSAN076620	99.7 (1/34)	99.6 (4/35)	99.8 (1/31)	99.5 (8/66)	4639
CFSAN086181	99.6 (2/21)	99.8 (0/10)	99.7 (1/15)	99.7 (4/20)	4841

^a^ Total genes in each wgMLST scheme (reference genome specific) used for the analysis.

**Table 6 ijms-25-11877-t006:** SNP matrix of the second wgMLST analysis for CFSAN000189, showing SNP differences among the different combinations of ONT sequencing and the PacBio and Hybrid assemblies.

		1	2	3	4	5	6	7	8	9
1	CFSAN000189_FAW25027_f	0	119	133	131	100	100	100	100	100
2	CFSAN000189_FAW42567_f	119	0	143	137	109	109	109	109	109
3	CFSAN000189_FAW43075_f	133	143	0	126	124	124	124	124	124
4	CFSAN000189_FAW18242_f	131	137	126	0	117	117	117	117	117
5	CFSAN000189 CP045753	100	109	124	117	0	0	0	0	0
6	CFSAN000189_M1	100	109	124	117	0	0	0	0	0
7	CFSAN000189_M2	100	109	124	117	0	0	0	0	0
8	CFSAN000189_M3	100	109	124	117	0	0	0	0	0
9	CFSAN000189_NCBI	100	109	124	117	0	0	0	0	0

NCBI—PacBio, CP045753—Hybrid ONT + Illumina, M—MiSeq.

**Table 7 ijms-25-11877-t007:** SNP matrix of the second wgMLST analysis for CFSAN123154, showing SNP differences among the different combinations of ONT sequencing and the PacBio assembly.

		1	2	3	4	5	6	7	8
1	CFSAN123154_FAW25027_f	0	159	101	102	101	101	138	153
2	CFSAN123154_FAW18242_f	159	0	130	131	130	130	160	171
3	CFSAN123154_M2	101	130	0	1	0	2	112	136
4	CFSAN123154_M1	102	131	1	0	1	3	113	137
5	CFSAN123154_M3	101	130	0	1	0	2	112	136
6	CFSAN123154_P	101	130	2	3	2	0	112	136
7	CFSAN123154_FAW43075_f	138	160	112	113	112	112	0	131
8	CFSAN123154_FAW42567_f	153	171	136	137	136	136	131	0

P—PacBio, M—MiSeq.

**Table 8 ijms-25-11877-t008:** SNP matrix of the second wgMLST analysis for CFSAN030807, showing SNP differences among the different combinations of ONT sequencing and the PacBio assembly.

		1	2	3	4	5	6	7	8
1	CFSAN030807	0	2	0	2	0	0	0	0
2	CFSAN030807_FAW18242_f	2	0	2	4	2	2	2	2
3	CFSAN030807_FAW25027_f	0	2	0	2	0	0	0	0
4	CFSAN030807_FAW42567_f	2	4	2	0	2	2	2	2
5	CFSAN030807_FAW43075_f	0	2	0	2	0	0	0	0
6	CFSAN030807_M1	0	2	0	2	0	0	0	0
7	CFSAN030807_M3	0	2	0	2	0	0	0	0
8	CFSAN030807_P	0	2	0	2	0	0	0	0

P—PacBio, M—MiSeq.

**Table 9 ijms-25-11877-t009:** SNP matrix of the second wgMLST analysis for CFSAN076620, showing SNP differences among the different combinations of ONT sequencing and the PacBio assembly.

		1	2	3	4	5	6
1	CFSAN076620_FAW18242_f	0	28	36	28	30	35
2	CFSAN076620_M1	28	0	9	3	2	9
3	CFSAN076620_FAW42567_f	36	9	0	12	11	9
4	CFSAN076620_FAW25027_f	28	3	12	0	5	10
5	CFSAN076620_P	30	2	11	5	0	11
6	CFSAN076620_FAW43075_f	35	9	9	10	11	0

P—PacBio, M—MiSeq.

**Table 10 ijms-25-11877-t010:** SNP matrix of the second wgMLST analysis for CFSAN086181, showing SNP differences among the different combinations of ONT sequencing and the PacBio, Hybrid, and Illumina assemblies.

		1	2	3	4	5	6	7	8	9	10
1	LEMB18 NZ_CP045782.1_hyb	0	0	0	0	1	1	9	3	4	0
2	CFSAN086181_M1	0	0	0	0	1	1	9	3	4	0
3	CFSAN086181_M2	0	0	0	0	1	1	9	3	4	0
4	CFSAN086181_M3	0	0	0	0	1	1	9	3	4	0
5	45006_SRR9042857	1	1	1	1	0	2	10	4	5	1
6	CFSAN086181_FAW25027_f	1	1	1	1	2	0	8	4	5	1
7	CFSAN086181_FAW18242_f	9	9	9	9	10	8	0	10	11	9
8	CFSAN086181_FAW42567_f	3	3	3	3	4	4	10	0	5	3
9	CFSAN086181_FAW43075_f	4	4	4	4	5	5	11	5	0	4
10	CFSAN086181_P	0	0	0	0	1	1	9	3	4	0

P—PacBio, M—MiSeq, hyb—Hybrid ONT + Illumina.

## Data Availability

Data is contained within the article and Appendix A.

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
