# Peer review of "Single Laboratory Evaluation of the Q20+ Nanopore Sequencing Kit for Bacterial Outbreak Investigations"

_ijms, 2024, doi:10.3390/ijms252211877_

Round 1

Reviewer 1 Report

Comments and Suggestions for Authors

The manuscript titled “ Single laboratory evaluation of the Q20+ nanopore sequencing kit for bacterial outbreak investigations” by Hoffmann, evaluated Oxford Nanopore Technologies (ONT) Q20+ chemistry for rapid, cost-effective pathogen identification in foodborne outbreaks, focusing on Shiga toxin-producing Escherichia coli (STEC) for leafy greens. By testing different DNA extraction and library preparation methods, the study  found that ONT Q20+ produced high-quality bacterial genomes and performed comparably to Illumina sequencing in species identification, genotyping, and virulence factor detection. However, variations in SNP accuracy and assembly completeness suggest further optimization is needed before broad application in outbreak surveillance. This is a novel aspect in a field of intense research. Still some issues need to be clarified, as listed below, before the manuscript can be accepted for publication in IJMS.

11.      For abstract part, “ This pilot study evaluated different combination of two DNA extraction methods( Maxwell RSC Cultured Cell DNA kit, and Monarch high molecular weight extraction kit) and two ONT library preparation protocols (ligation and the rapid barcoding sequencing kit) using five well-characterized strains representing diverse foodborne pathogens.” is repeated twice. “ This study aimed to evaluate the potential Oxford Nanopore Technologies (ONT) with Q20+ chemistry as a cheaper, rapid and accurate method for identifying and clustering foodborne pathogens. The study focuses on assessing whether ONT Q20+ technology could offer near real-time pathogen identification, including SNP differences, serotypes, and antimicrobial resistance genes, to overcome the drawbacks of existing methodologies.” also repeated twice. Before submitting your final draft, I strongly recommend carefully reviewing the manuscript to avoid such repetitive content other minor errors. Ensuing clarify and avoiding redundancy will strengthen the overall quality of your submission. Please rewrite the abstract to maintain focus and conciseness.

22.       The study employed two different DNA extraction kits (Maxwell RSC Cultured Cell DNA kit and Monarch HMW DNA extraction kit) Could you please elaborate on how the differences in DNA quality produced by these kits impacted the assembly accuracy and completeness of the bacterial genomes? Additionally, it would be helpful to understand whether any biases were introduced during the extraction and how these may have influenced the results.

33.      In your analysis, the ONT assemblies showed SNPs and indels compared to the reference genomes, with various across different species. Could you clarify how these differences were accounted for when conducting the phylogenomic clustering? Additionally, could these SNPs and indels potentially impact the accuracy of pathogen identification in real-world outbreak investigations?

44.      This study was conducted using five specific bacterial strains. How generalizable are the findings to other strains, especially those with varying GC content or more complex genomes? Additionally, do you have plans to expand the study to include a broader range of pathogens to more thoroughly assess the robustness of ONT Q20+ chemistry for diverse foodborne pathogens?

Author Response

 Comments 1: The manuscript titled “ Single laboratory evaluation of the Q20+ nanopore sequencing kit for bacterial outbreak investigations” by Hoffmann, evaluated Oxford Nanopore Technologies (ONT) Q20+ chemistry for rapid, cost-effective pathogen identification in foodborne outbreaks, focusing on Shiga toxin-producing Escherichia coli (STEC) for leafy greens. By testing different DNA extraction and library preparation methods, the study  found that ONT Q20+ produced high-quality bacterial genomes and performed comparably to Illumina sequencing in species identification, genotyping, and virulence factor detection. However, variations in SNP accuracy and assembly completeness suggest further optimization is needed before broad application in outbreak surveillance. This is a novel aspect in a field of intense research. Still some issues need to be clarified, as listed below, before the manuscript can be accepted for publication in IJMS.

Response 1: We thank the reviewer for their comments that have improved our publication. Overall, we believe the revised manuscript has addressed the reviewers’ comments and suggestions, resulting in a stronger and more comprehensive presentation of our findings.

Comments 2: For abstract part, “ This pilot study evaluated different combination of two DNA extraction methods( Maxwell RSC Cultured Cell DNA kit, and Monarch high molecular weight extraction kit) and two ONT library preparation protocols (ligation and the rapid barcoding sequencing kit) using five well-characterized strains representing diverse foodborne pathogens.” is repeated twice. “ This study aimed to evaluate the potential Oxford Nanopore Technologies (ONT) with Q20+ chemistry as a cheaper, rapid and accurate method for identifying and clustering foodborne pathogens. The study focuses on assessing whether ONT Q20+ technology could offer near real-time pathogen identification, including SNP differences, serotypes, and antimicrobial resistance genes, to overcome the drawbacks of existing methodologies.” also repeated twice. Before submitting your final draft, I strongly recommend carefully reviewing the manuscript to avoid such repetitive content other minor errors. Ensuing clarify and avoiding redundancy will strengthen the overall quality of your submission. Please rewrite the abstract to maintain focus and conciseness.

Response 2: For the abstract, the repeated sentences have been removed. We have conducted a thorough review to avoid such repetitive content in the final draft.

Comments 3: The study employed two different DNA extraction kits (Maxwell RSC Cultured Cell DNA kit and Monarch HMW DNA extraction kit) Could you please elaborate on how the differences in DNA quality produced by these kits impacted the assembly accuracy and completeness of the bacterial genomes? Additionally, it would be helpful to understand whether any biases were introduced during the extraction and how these may have influenced the results.

Response 3:  Regarding the differences between the DNA extraction kits, we did not observe any significant impact on the assembly accuracy or completeness of the bacterial genome. The quantity and quality of the DNA were similar between the two extraction methods. The main difference was the integrity of the DNA and the resulting fragment lengths, which did not affect our assembly results.

Comments 4: In your analysis, the ONT assemblies showed SNPs and indels compared to the reference genomes, with various across different species. Could you clarify how these differences were accounted for when conducting the phylogenomic clustering? Additionally, could these SNPs and indels potentially impact the accuracy of pathogen identification in real-world outbreak investigations?

Response 4: For the SNPs and Indels observed in the ONT assemblies, we clarified that indels were not included in the phylogenetic analyses, so they did not affect the phylogenetic clustering. Only shared SNP positions by everyone were used in the wgMLST and final phylogenetic tree but would not affect pathogen identification during real-world outbreak investigations. We have clarified this in the “Phylogenetic relationship of the strains determined by wgMLST analysis” section as follows: “After eliminating any loci that were missing or found containing indels (showed as failed loci in the software) from the genome of any strain used in our analyses, we performed the wgMLST analysis”.

Comments 5:  This study was conducted using five specific bacterial strains. How generalizable are the findings to other strains, especially those with varying GC content or more complex genomes? Additionally, do you have plans to expand the study to include a broader range of pathogens to more thoroughly assess the robustness of ONT Q20+ chemistry for diverse foodborne pathogens?

Response 5:  To address the generalizability of the findings, we have conducted a follow-up study examining a broader set of 32 bacterial strain, including Gram-positive and Gram-negative organisms with varying GC content. You can find those strains under bioproject number: PRJNA1101017. They have been also sequenced with Illumina as well as PacBio for comparison purposes. The results show that the performance of the ONT Q20+ chemistry varies across different species and outbreak scenarios, but. It can still provide accurate clustering, even if the branch lengths may differ compared to Illumina data. We plan to further expand the evaluation to include a more diverse range of foodborne pathogens.

Reviewer 2 Report

Comments and Suggestions for Authors

The study compared sequencing technologies and their effectiveness in identifying enteropathogenic E. coli strains and the potential to discover antibiotic resistance determinants. Although the article contains a great deal of experimental material, it must be corrected before publication.

First of all, scientific meaning must be given to the experiments made. Is this the first time Oxford Nanopore Technologies has been applied to these pathogens? What application will this have for medicine and public health?

1. The abstract does not meet the requirements of the journal for several reasons:

- it's too long

- it contains repetitions of entire passages. For example, the sentence "The study focuses on assessing whether ONT Q20+ technology could offer ..." and a whole paragraph after it is repeated twice

- it sounds like an advertising brochure (for example the sentence "offering valuable insights into the practical application of nanopore sequencing"), but not like a synthesized result of scientific research.

The abstract must be completely rewritten.

2. Introduction:

- The authors should describe the principle of Oxford Nanopore Technology

- As you examine the qualities of technology, you can put a workflow for the study, which includes a flow chart for the proposed comparison of the two ways to isolate DNA and to prepare a library.

3. The introduction should explain why you chose these particular pathogen species and these particular strains. Are they associated with outbreaks? Are they monitored by health services?

3. Results:

Table 3 needs to be reformatted to make it readable.

4. The discussion is weak and short. It does not make it clear which of the DNA isolation methods was better, or which method of library creation was better. The authors should discuss the meaning of the work and underline the novelty. More accurate sequence or faster, if cheaper - why?

5. Over 23.5% of the references are self-citations, this is too much and must be reduced.

Author Response

Comments 1: The study compared sequencing technologies and their effectiveness in identifying enteropathogenic E. coli strains and the potential to discover antibiotic resistance determinants. Although the article contains a great deal of experimental material, it must be corrected before publication.

Response 1: We appreciate the reviewer’s suggestions. We have addressed all the comments in the revised manuscript.

Comments 2: First of all, scientific meaning must be given to the experiments made. Is this the first time Oxford Nanopore Technologies has been applied to these pathogens? What application will this have for medicine and public health?

Response 2:  No, this is not the first time that ONT has been applied to these pathogens. These are among the most common foodborne pathogens causing foodborne outbreaks worldwide. Foodborne pathogen surveillance in the majority of developed countries is done using genomics. Until recently most genomic surveillance was done using short read sequencing (mostly Illumina sequencing). ONT until recently because of their lower accuracy was used mostly for supporting Illumina sequencing to close bacterial genomes (besides foodborne pathogens). However, since the release of the R10.4.1 and 114 sequencing kits, the accuracy has increased dramatically to be almost on par with Illumina accuracy (>Q40). Our study aimed to test the performance of this newer ONT release for foodborne pathogen sequencing. The relevance to medicine and public health is that it can provide long-read data sequencing in real-time, that can be analyzed and provide fast pathogen identification during outbreaks (e.g., foodborne illnesses, hospital-acquired infections, pandemics).  Furthermore, ONT sequencing can rapidly detect genes associated with antimicrobial resistance, helping guide treatment decisions and monitor the spread of resistance in hospital settings.    

Comments 3: The abstract does not meet the requirements of the journal for several reasons:

- it's too long, - it contains repetitions of entire passages. For example, the sentence "The study focuses on assessing whether ONT Q20+ technology could offer ..." and a whole paragraph after it is repeated twice,  - it sounds like an advertising brochure (for example the sentence "offering valuable insights into the practical application of nanopore sequencing"), but not like a synthesized result of scientific research. The abstract must be completely rewritten.

Response 3:  Agreed, the abstract has been completely rewritten to improve focus and conciseness, while avoiding repetition.

Comments 4: Introduction: - The authors should describe the principle of Oxford Nanopore Technology

Response 4:  In the introduction, we have added information about the principles of Oxford. We have added the following in the introduction: “An alternative approach involves using Oxford Nanopore Technologies (ONT) sequencing, a technology that allows for determining a DNA or RNA sequence by reading DNA or RNA through a biological nanopore and measuring changes in an electrical current that translates into a base call. ONT produces long reads which can identify taxa while sequencing”

Comments 5: - As you examine the qualities of technology, you can put a workflow for the study, which includes a flow chart for the proposed comparison of the two ways to isolate DNA and to prepare a library.

Response 5: Thank you for your valuable suggestion. We agree that a workflow diagram would complement the information already presented in the table 2 and provide a clearer visual representation of the comparison between the two methods for DNA isolation and library preparation. To address this, we have included a workflow diagram to illustrate the comparison of the DNA extraction and library preparation methods in supplementary materials as supplementary figure 1.

Comments 6: The introduction should explain why you chose these particular pathogen species and these particular strains. Are they associated with outbreaks? Are they monitored by health services?

Response 6: Agreed and done. We have provided more context on why the specific pathogen species and strains were selected, noting their significance in terms of public health and outbreaks.  We added an extra sentence:” For this pilot study we selected to sequence five well known and characterized foodborne and healthcare-associated bacterial pathogen strains that pose significant public health threats, belonging to five different species from our collection: Salmonella enterica subsp. enterica, Vibrio parahemolyticus, Shigella sonnei, Escherichia coli, and Klebsiella pneumoniae (Table 1). These organisms are closely monitored by health services due to their roles in outbreaks, potential for severe illness, and emerging antibiotic resistance.”

Comments 7: 3. Results: Table 3 needs to be reformatted to make it readable.

Response 7: Agreed and done. Table 3 has been reformatted to improve readability.

Comments 8: The discussion is weak and short. It does not make it clear which of the DNA isolation methods was better, or which method of library creation was better. The authors should discuss the meaning of the work and underline the novelty. More accurate sequence or faster, if cheaper - why?

Response 8: The discussion section has been expanded to provide a clearer comparison of the DNA isolation and library preparation methods, highlighting their strengths and limitations. We have also emphasized the novelty of our study in evaluating the performance of the ONT Q20+ chemistry across multiple bacterial species and sequencing runs.

Comments 9: Over 23.5% of the references are self-citations, this is too much and must be reduced.

Response 9:  Thank you for your observation regarding the self-citations. While we acknowledge that 24% of the references are self-citations, we believe that these references are critical to providing a comprehensive background and context for the current study. These prior publications contain foundational work and data that are directly relevant to the methods and findings presented here. That said, we understand the concern, and we have added additional publications that are necessary for supporting the content.

Reviewer 3 Report

Comments and Suggestions for Authors

The manuscript, titled "Single laboratory evaluation of the Q20+ nanopore sequencing kit for bacterial outbreak investigations," provides a detailed analysis of nanopore sequencing technology for rapidly identifying pathogens. The study has significant potential for improving foodborne pathogen detection, and its findings are timely, given the need for more efficient outbreak response protocols.

1. The authors must address the discrepancies observed in genome completeness and assembly length for specific strains. Whether these differences arise from the DNA extraction methods or the library preparation process is unclear. To explain these inconsistencies, further clarification is required, especially for the Shigella sonnei and Klebsiella pneumoniae strains.

2. The comparison between ONT Q20+ and Illumina sequencing is appropriate, but the manuscript would benefit from a deeper analysis of previous studies that also used nanopore sequencing, particularly in outbreak investigations. Highlighting the advances of the Q20+ chemistry over earlier versions of ONT technology would strengthen the narrative and emphasize the improvements.

3. The persistence of SNP differences and indels, even after hybrid assemblies with Illumina data, requires more attention. These variations could have significant consequences in outbreak investigations, where high precision at the strain level is crucial. The authors should discuss the potential impact of these issues on real-world applications and consider whether additional steps, such as software optimization, could mitigate these problems.

4. The manuscript should offer a more straightforward explanation of how SNPs and indels influence the phylogenomic tree. For example, how might these variations affect the conclusions drawn during an outbreak investigation, particularly for strains with minimal SNP differences? This is important for understanding the potential limitations of the technology in high-stakes scenarios.

Author Response

Comments 1: The manuscript, titled "Single laboratory evaluation of the Q20+ nanopore sequencing kit for bacterial outbreak investigations," provides a detailed analysis of nanopore sequencing technology for rapidly identifying pathogens. The study has significant potential for improving foodborne pathogen detection, and its findings are timely, given the need for more efficient outbreak response protocols.

Response 1:  Thank you for your positive feedback and for recognizing the potential impact of our study. We appreciate your acknowledgment of the detailed analysis provided on nanopore sequencing technology for rapid identification of bacterial pathogens. We agree that the findings are timely and could contribute to more efficient outbreak response protocols, especially in the context of foodborne pathogen detection. We are confident that the improvements highlighted in the study will offer valuable insights for advancing pathogen surveillance and control efforts.

Comments 2:  The authors must address the discrepancies observed in genome completeness and assembly length for specific strains. Whether these differences arise from the DNA extraction methods or the library preparation process is unclear. To explain these inconsistencies, further clarification is required, especially for the Shigella sonnei and Klebsiella pneumoniae strains.

Response 2:  Thank you for raising this point. We understand the concern regarding the differences in genome completeness and assembly length, particularly for the Shigella and Klebsiella pneumoniae strains. However, we believe these differences are not attributable to the DNA extraction methods or library preparation processes. In our experience, variation in assembly results can sometimes arise from the downstream analysis itself. Minor discrepancies in assembly length or genome completeness are common and can depend on the specific assembler used or the parameters applied during analysis. In the case of Shigella and Klebsiella, these genomes do not present the level of complexity (e.g., repetitive regions or significant structural variations) that would make them particularly sensitive to the DNA extraction or library preparation methods. Moreover, both extraction methods used in this study consistently achieved sufficient coverage (above 30X), which we believe is adequate to generate a complete or near-complete genome regardless of the library preparation method. The key difference in output between the rapid kit and other library preparation methods primarily relates to the total read yield. We have added the following in the discussion section:The differences observed in genome completeness and assembly length for Shigella and Klebsiella pneumoniae are likely due to variations in the downstream assembly process rather than the DNA extraction or library preparation methods. Given that both methods consistently achieved sufficient coverage (above 30X), we do not expect the extraction or preparation methods to significantly impact genome quality, as the primary difference between the methods relates to total read yield rather than assembly accuracy.”

Comments 3: The comparison between ONT Q20+ and Illumina sequencing is appropriate, but the manuscript would benefit from a deeper analysis of previous studies that also used nanopore sequencing, particularly in outbreak investigations. Highlighting the advances of the Q20+ chemistry over earlier versions of ONT technology would strengthen the narrative and emphasize the improvements.

Response 3: ONT has never been used as a standalone technology during a foodborne outbreak investigation. Until now it has been used in retrospective outbreak analysis mostly using reference mapping analysis methods. In general, ONT has been used together with Illumina data to produce high quality closed genomes. However, we agree with the reviewer that the Q20+ chemistry is dramatically more accurate than the previous chemistry and we have added the following sentence to the discussion to highlight the differences regarding the previous ONT chemistry:” A key innovation of this study is that it demonstrates the potential of ONT newer chemistry to deliver more accurate sequences than the previous ONT chemistry (R9.4.1), faster, and at a lower initial cost, making it a more efficient tool for bacterial screening during outbreak investigations.   “

Comments 4: The persistence of SNP differences and indels, even after hybrid assemblies with Illumina data, requires more attention. These variations could have significant consequences in outbreak investigations, where high precision at the strain level is crucial. The authors should discuss the potential impact of these issues on real-world applications and consider whether additional steps, such as software optimization, could mitigate these problems.

 Response 4:  Thank you for your comment. We did not observe indels during hybrid assemblies, and most had zero SNPs when compared to the reference genome. While SNP variation is a key consideration in outbreak investigations, it is not uncommon to observe genomes differing by several SNPs, which can still be part of the same outbreak due to natural mutation rates during strain culturing. It is also important to emphasize that SNPs are not the sole determining factor in outbreak inclusion; epidemiological data plays a critical role in confirming relatedness between strains. We have added the following to the text to address this comment: “While SNP variation is important, genomes within an outbreak may differ by a few SNPs due to natural mutation rates during strain culturing. Therefore, by integrating genetic data with epidemiological context is essential for accurately interpreting phylogenetic relationships and assessing potential outbreak inclusion, ultimately enhancing our ability to draw informed conclusions

Comments 5: The manuscript should offer a more straightforward explanation of how SNPs and indels influence the phylogenomic tree. For example, how might these variations affect the conclusions drawn during an outbreak investigation, particularly for strains with minimal SNP differences? This is important for understanding the potential limitations of the technology in high-stakes scenarios.

Response 5:  Thank you for your insightful comment. We appreciate the opportunity to clarify the influence of house SNPs and indel variations on phylogenetic trees. As we mentioned previously, indels are excluded from our phylogenetic analysis, ensuring that only shared positions among all strains are considered. This approach helps maintain the integrity of the tree structure, as it reflects the most reliable genetic information. Furthermore, while SNP variations are indeed critical in assessing relationships among strains, it is important to recognize that some SNP differences can occur naturally due to mutation rates during strain culturing. Therefore, it is not uncommon for genomes within the same outbreak to exhibit a few SNP differences. This reality underscores the necessity of incorporating epidemiological data alongside genetic analysis to make informed decisions about outbreak inclusion. In high-stakes scenarios, understanding these variations and their implications is vital for accurately interpreting phylogenetic results. By combining genetic insights with epidemiological context, we can better assess the potential limitations of the technology and ensure robust conclusions regarding outbreak dynamics. We have added the following to our conclusions: “While SNP variation is important, genomes within an outbreak may differ by a few SNPs due to natural mutation rates during strain culturing. Therefore, by integrating genetic data with epidemiological context is essential for accurately interpreting phylogenetic relationships and assessing potential outbreak inclusion, ultimately enhancing our ability to draw informed conclusions”

Round 2

Reviewer 2 Report

Comments and Suggestions for Authors

The authors have complied with all comments and answered all questions in detail and comprehensively. The corrections are substantial and have greatly improved the meaning of the exposition.

The manuscript may be accepted for publication in its current form.

Reviewer 3 Report

Comments and Suggestions for Authors

The manuscript has been revised thoroughly in response to the previous comments.

The authors have addressed all concerns effectively, and the quality of the manuscript has improved significantly.

I find the current version suitable for publication in its present form.